# Community-identified priorities for parenting and parental mental health in coastal Ghana: Formative insights to shape a multi-component intervention

Faiza Abdul[1], Kafui K. Dzorgbesi[2], Shirley-Anne Lutterodt[2], David D. Kotey[2], Richard Appiah[2,3,4], Marilyn Naana Ahun[1,2,5]*

1 Centre for Outcomes Research and Evaluation, McGill University Health Centre – Research Institute, Montréal, Quebec, Canada, 2 Africa Centre for Well-Being Research and Advocacy (Africa-Well), Accra, Ghana, 3 College of Health Sciences, University of Ghana, Accra, Ghana, 4 School of Psychology, Northumbria University, Newcastle-upon-Tyne, United Kingdom, 5 Department of Medicine, Faculty of Medicine and Health Sciences, McGill University, Montréal, Quebec, Canada

* marilyn.ahun@mcgill.ca

## Abstract

Interventions that combine parenting support with parental mental health components hold promise for enhancing child and family well-being. Yet, few integrate robust mental health strategies or draw on formative research to ensure contextual and cultural relevance. We conducted qualitative research in Ghana to explore local experiences of parenting and parental mental health to inform the development of a multi-component intervention. We conducted 31 in-depth interviews and 6 focus group discussions with 83 parents, nonparental caregivers, and community members across three districts in southern Ghana. Reflexive thematic analysis was used to identify key patterns. Participants highlighted facilitators and barriers to engagement in positive parenting practices and supporting mental health. Maintaining a positive couple's relationship, financial stability, awareness of appropriate disciplinary methods, and participation in religious and social gatherings facilitated positive parenting. Barriers included financial difficulties, limited time with children due to work, and lack of paternal involvement. Supporting parental mental health was facilitated by religious engagement, financial stability, and strong social support, whereas financial difficulties, social isolation, rigid gender norms, and lack of access to mental health services were barriers. All parents and nonparental caregivers expressed interest in attending a parenting and mental health support program, highlighting the need for community-based support. Our results highlight the need for culturally relevant multi-component strategies that reflect local needs. Findings will inform the design of a multi-component parenting and parental mental health intervention to improve child and family well-being.

**Data availability statement:** The data analyzed during this study are included in this article. As our data were collected using qualitative research methods from a relatively small group of participants, data cannot be made available to third parties as there is a risk that study participants may be identifiable.

**Funding:** This work was supported by the Society for Research in Child Development (Victoria S. Levin Award for Early Career Success in Young Children's Mental Health Research to MNA). The funders had no role in study design, data collection and analysis, decision to publish, or preparation of the manuscript.

**Competing interests:** The authors have declared that no competing interests exist.

## Introduction

Globally, nearly half of children under five years in low- and middle-income countries are at risk of not reaching their full developmental potential due to preventable social and environmental factors [1]. In Ghana, a lower-middle-income country in West Africa, this figure ranges from 40–59%, underscoring the urgency for early interventions [2]. The first few years of a child's life are critical for long-term development, and factors such as poverty, conflict within the home, parental mental health difficulties, and suboptimal parenting can negatively shape early childhood development (ECD) [1]. Parenting behaviours and parental mental health in particular are two critical early childhood factors that are closely linked and jointly influence ECD [1]. Caregivers experiencing psychological distress, particularly depression, often struggle to provide responsive, emotionally-attuned care, which can undermine children's developmental outcomes [3,4]. While standalone parenting or mental health interventions yield modest gains, evidence suggests they rarely produce sustained, holistic improvements in both caregiver and child well-being [5–8].

Although multi-component interventions promise improvements in both parenting and parental mental health, reviews of interventions from both high- and low-resource contexts reveal that most fall short, often due to an overemphasis on parenting strategies with limited attention to mental health [9–11]. This is evident in the findings of a meta-analysis on the impact of multi-component interventions on caregiver outcomes, which found no significant impact on reducing caregiver depressive symptoms [9]. Furthermore, only a small handful of interventions assessed parenting practices, preventing a meta-analysis of intervention impact on this outcome. A closer look at the content of these multi-component interventions reveals that those with only one or two sessions focused on parental mental health had limited impacts on improving both caregiver and child mental health outcomes [12,13]. Existing approaches that simply 'add on' one or two mental health sessions to parenting curricula (or vice versa) represent tokenistic integration rather than systematic design informed by prior evidence and caregiver needs. This superficial integration likely explains their limited effectiveness.

Another important limitation of prior interventions is that few are grounded in formative research that explores how interventions can be tailored to local realities [14]. Formative research consists of gathering information on relevant behaviours in a given context, including if and how those behaviours are practiced, barriers and facilitators to engaging in those behaviours, and the existence and implementation of services targeting the behaviours [15]. Such work is vital in the development of any intervention, and failure to engage in this process of contextual adaptation risks cultural misalignment and reduces effectiveness. For example, a US-based multi-component intervention trialled in northern Ghana failed to improve maternal depression or children's socio-emotional development, likely due to its lack of contextual relevance [16]. Specifically, the intervention only targeted mothers, presuming that they were children's sole caregivers. In collectivistic contexts such as Ghana,

omitting fathers and nonparental caregivers overlooks key actors in children's daily environment and their role in influencing child development [2,17].

In light of these gaps, this study aims to generate formative evidence on parenting and parental mental health in Ghana to guide the development of a contextually-relevant multi-component intervention that addresses families' needs. Specifically, we aim to (1) understand how parents and nonparental caregivers care for their young children and support their own mental health, (2) identify the factors that facilitate or hinder their engagement in these behaviours, and (3) examine community services to support parents and nonparental caregivers. We use the term "caregivers" to refer to both parents and nonparental caregivers.

## Methods

### Ethics statement

The study protocol was reviewed and approved by the Institutional Review Boards of the Ghana Health Service (012/04/24) and McGill University Health Centre – Research Institute (2024-10502). Informed consent was obtained orally and in writing, with forms read aloud in local languages to accommodate varying literacy levels. Participants were assured of confidentiality and their right to withdraw at any point. Participants were given bars of soap as a token of appreciation for their time. Refreshments were also provided for focus group discussion (FGD) participants given the length of discussions.

### Study design

We adopted a descriptive phenomenological design [18] to elicit in-depth accounts of caregiving roles and mental health difficulties from those with lived experience. This approach was well-suited to understanding the subjective meaning caregivers assign to their everyday experiences. We conducted individual in-depth interviews (IDIs) with mothers, fathers, nonparental caregivers of young children, and community members (e.g., health workers, religious leaders, community elders) along with FGDs with mothers and fathers. The COREQ checklist guided the reporting of the methodology and study results (S1 File) [19].

### Study site

The study took place in Kokrobite, Tuba, and Bortianor, fishing communities in Ghana's Ga South district, where gendered labour roles and socio-economic constraints shape family life [20–22].This site was chosen due to its sociodemographic similarities with other coastal towns and the authors' established relationships with community elders through previous research [20,23]. These peri-urban areas report high rates of socioemotional difficulties among children (47%) and significant parental mental health difficulties, with maternal and paternal depression estimated at 35% and 26%, respectively [23].

### Sampling

A purposive sampling strategy was employed to ensure representation of diverse caregiving perspectives across various sociodemographic characteristics (e.g., gender, age, educational attainment, socio-economic status, etc.). We engaged three community mobilizers, identified by local assembly and health officials, to recruit participants based on their familiarity with household caregiving dynamics in study areas. Mobilizers were trained with and followed a standardized recruitment guide that outlined the study's eligibility criteria, communication protocols, and procedures for approaching potential participants [24]. The project coordinator (DDK) met periodically with mobilizers throughout recruitment to ensure adherence to the guide and diversity of recruited participants. Eligible participants included mothers and fathers (≥ 18 years) who were primary caregivers of children aged 6–60 months and lived together in the same household with the child.

Additional participants included nonparental caregivers (≥ 18 years) who provided routine, unpaid care for 6–60-month-old children and respected community members (≥ 18 years) such as religious leaders, community elders, and health workers. Including parents, nonparental caregivers, and community leaders enabled triangulation across caregiving roles and social perspectives, enhancing the credibility and depth of findings. We recruited caregivers of children 6–60 months because the first five years of life are a critical period for brain development, where children are particularly sensitive to their environment [1].

## Data collection

Out of 91 people contacted, 83 agreed to participate. The primary reason for refusal was survey fatigue. The sample size was based on other qualitative studies of a similar nature and adapted during data collection through the monitoring of results to assess whether findings had reached a point of saturation [25–27]. We conducted 31 IDIs with fathers, mothers, nonparental caregivers, and community members and 6 FGDs with 52 participants across father-only, mother-only, and mixed-parent groups (Table 1). On average, IDIs lasted 48 minutes and FGDs lasted 1 hour 51 minutes. Data collection took place between August 28 and September 8, 2024, using semi-structured interview guides designed to explore parenting roles and behaviours, parental support systems, and parental mental health. The senior author led the development of study tools, creating four separate topic guides for IDIs and FGDs for each respondent group. These guides were refined during training and pilot interviews with four mothers, two nonparental caregivers, and one community member in study areas. Each FGD included 8–11 participants (mothers or fathers) who took part in structured discussions aligned with the guides' thematic areas. IDIs and FGDs were conducted concurrently in quiet and private locations in each community and audio recorded. All transcripts were included in the analysis to ensure a comprehensive understanding of caregiving and parental mental health experiences.

To ensure rigorous data collection, nine research assistants were recruited from the Africa Centre for Well-Being Research and Advocacy's database based on education level, previous qualitative research experience, and local language proficiency. The principal investigator (MNA) and project coordinator led a five-day training covering study aims, interviewing techniques, transcription standards, and research ethics. Emphasis was placed on culturally sensitive translation of key concepts into Ga-Dangme, supported by role plays and pilot interviews. Following assessments, four trainees were selected as interviewers (including KKD and SL), one was placed on standby, and four were assigned as transcribers. Each transcriber was paired with an interviewer, and a shared folder facilitated secure audio file transfers. Transcripts and field notes were uploaded at the end of each day of data collection, translated, and double-checked by the project coordinator, a native Ga-Dangme speaker, to ensure accuracy and minimize errors.

**Table 1. Number of participants in IDIs and FGDs.**

| Data collection method | Respondent type | Number of participants |
| --- | --- | --- |
| IDI | Father | 10 |
| | Mother | 10 |
| | Nonparental caregiver | 5 |
| | Community member | 6 |
| FGD | Father | 16[a] |
| | Mother | 16[a] |
| | Mixed (mother and father) | 20[b] |
| TOTAL | | 83 |

[a] Each FGD consisted of 8 participants.

[b] One FGD consisted of 9 participants (4 fathers, 5 mothers) and the other had 11 (4 fathers, 7 mothers).

## Data analysis

Data were analyzed using reflexive thematic analysis [28,29] and in line with a framework that sought to summarize how caregivers cared for their young children and their own mental health and determine the main barriers and facilitators to engaging in these tasks. Two analysts (FA, MNA) developed a preliminary codebook informed by study aims, topic guides, field reflections, and relevant literature [25–27]. The codebook was piloted and refined iteratively before applying it across all transcripts using line-by-line coding and documented memos in Atlas.ti. Weekly meetings were held from October 2024-January 2025 throughout the coding process to review memos, resolve any uncertainties, and discuss emerging themes and cultural and contextual nuances. Once coding was complete, a team of four analysts (FA, KKD, SL, MNA) collaboratively synthesized the codes and had weekly meetings in February 2025 to consolidate themes and draft result summaries. Results were triangulated across the perspectives of various respondent groups and different interview methods.

## Team reflexivity

This study is led by two Ghanaian researchers (MNA, RA) with expertise in mental health, parenting, and qualitative research in Ghana. Both researchers have established relationships with community elders in the study sites and have previously engaged in research and dissemination activities with community members [23]. Data were collected by a team of four (one male, three females) trained Ghanaian research assistants, supervised by a male Ghanaian project coordinator (DDK), with bachelor and graduate degrees and extensive field experience conducting mental health and parenting research in the study sites and elsewhere in Ghana. They were all either native speakers or fluent in the local languages of study areas. Two research assistants (KKD, SL) also participated in data analysis alongside the first (a Canadian-Pakistani psychology graduate with experience in qualitative research) and senior authors. The analysis team included members with lived caregiving and mental health experiences, which enhanced sensitivity to participants' narratives. Familiarity with the cultural context and local languages supported culturally nuanced interpretation and reduced outsider bias (S2 File).

## Results

This study included 83 participants. Most were self-employed (77%), and nearly half of the parents had some secondary education. Children's mean age was 32 months (range 6–60), with 60% enrolled in preschool. More information about participants' sociodemographic characteristics can be found in Table 2.

## Themes

Findings are organized into four broad themes (and their sub-themes) in line with our specific aims: (1) overall parenting experiences, (2) positive parenting behaviours, (3) parental mental health, and (4) existing community services. Sub-themes on current practices within themes 2 (positive parenting behaviours) and 3 (parental mental health) address our first specific aim by describing how caregivers care for their young children and support their own mental health. Within each of these themes, participants also discuss the specific barriers and facilitators caregivers face when engaging in each of these behaviours, shedding light on specific aim 2. Additionally, theme 4 allows an examination of community services to support caregivers, as well as caregivers' experiences accessing them (specific aim 3). The first theme on overall parenting experiences provides helpful context within which to interpret all other themes. These themes and sub-themes are summarized in Table 3.

**Parenting experiences: Gendered parenting roles.** Participants' accounts revealed a deeply gendered division of parenting roles, with fathers cast as financial providers and mothers as primary caregivers, a pattern often justified through religious and cultural norms yet occasionally contested. For example, some fathers acknowledged that mothers could contribute financially if the father was not earning enough money or if the father was absent. Others reported

**Table 2. Sample characteristics.**

| Respondents | *N* (%) or mean ±SD |
|---|---|
| *Caregivers* | N = 77 |
| Relationship to the child | |
| Mother | 38 (49%) |
| Father | 34 (44%) |
| Other (aunt, grandmother) | 5 (7%) |
| Age (in years) | 33.57 ± 8.7 |
| Highest education level | |
| None | 8 (10%) |
| Primary | 24 (31%) |
| Secondary: Junior High | 31 (40%) |
| Secondary: Senior High | 12 (16%) |
| Tertiary | 2 (3%) |
| Occupation | |
| Salaried | 5 (6%) |
| Self-employed | 59 (77%) |
| Apprenticement | 4 (5%) |
| Unemployed | 9 (12%) |
| Child age (in months) | 32 ± 17.8 |
| Child sex | |
| Male | 40 (51%) |
| Female | 38 (49%) |
| Education[a] | |
| None | 26 (36%) |
| Creche | 13 (18%) |
| Nursery | 14 (19%) |
| Kindergarten | 16 (22%) |
| Class 1 | 3 (4%) |
| *Respected community members* | N = 6 |
| Age (in years) | 55.83 ± 20.2 |
| Sex | |
| Male | 2 (33%) |
| Female | 4 (67%) |
| Community member role | |
| Religious leader (pastor, imam) | 2 (33%) |
| Healthcare personnel | 2 (33%) |
| Linguist | 1 (17%) |
| Assemblyman | 1 (17%) |
| Highest education level | |
| None | 1 (17%) |
| Primary | 2 (33%) |
| Tertiary | 3 (50%) |

[a] Data on child education were only available for n = 72 children.

**Table 3. Major themes and sub-themes.**

| Themes | Sub-themes | Example quotes |
|---|---|---|
| Parenting experiences | | *"A father can take care of the child, but you can't be compared to the mother, do you get me. That's why they say sweet mother [referring to a well-known song], do you get me, uh huh."* (Father-IDI-#1 |
| Positive parenting behaviours | Practices | *"I give them house training. And because I didn't go to school the house training my father taught me is what once a while I remember...like, uh huh, for instance like advising you not to join any bad company, joining gang in school."* (Father-IDI-#6). |
| | Facilitators | *"It's like the church does that in terms of programs and then when people come to church on Sunday, we try to educate them about most of the social challenges..."* (Community member-IDI-#8) |
| | Barriers | *"I do remember, my daughter, my first born she wanted to tell us her aspiration but because the father doesn't stay at home she can't tell the father. So, she wrote it on a paper and posted in the room, when you enter the room, you will see it posted there, she wrote and pasted it on the wall."* (Mixed FGD-#13). |
| Parental mental health | Practices | *"I try not to rely on anybody. I pray to God, sometimes he uses people to help me. That sometimes makes the sadness go away and I'm happy."* (Nonparental caregiver-IDI-#16) |
| | Facilitators | *"...But if I work and earn something, I feel happy that whenever any child of mine will face any problem, maybe in terms of an illness or the likes, I have something on me and can solve it for them."* (Father-IDI-#18 |
| | Barriers | *"There are a lot of needs, that I can be up from morning, but I do not have money to use to take care of the family, then it becomes a major problem for me. It becomes a thinking problem for me. Then if for twenty-four hours, I do not get any help from anywhere, then this problem will be on my mind."* (Father-IDI-#18) |
| Existing community services | Experiences | *"At times nurses are mobilized to come round the community to talk to us and educate us on how to take care of our children to avoid contracting diseases. There are also two polyclinics here where we go to whenever we feel sick."* (Mother-FGD-#27). |
| | Interest in additional programs | *"Yes, it depends on how you send the message, who gives the message and where you give the message. Always involve community leaders."* (Community member-IDI-#8) |

that while they helped their partner with childcare, they described their role as being incomparable or secondary to the mother's: *"A father can take care of the child, but you can't be compared to the mother, do you get me. That's why they say sweet mother [referring to a well-known song], do you get me, uh huh." (Father-IDI-#1)*.

Some fathers believed it important for men to engage in chores and childcare to strengthen relationships, while others argued that societal expectations and potential ridicule discouraged men from engaging in such activities. Specifically, fathers talked about being mocked by people in the community because they engage in household chores: *"You know, what my sister said, for instance, whenever men help them other people see that and they gossip behind them that the husband washes his wife's [clothes]. A friend of mine was doing that later on they were insulting him with that." (Mixed FGD-#2)*. Some fathers expressed a contrasting view, highlighting that sharing responsibilities helps promote unity and prevent issues such as family separation:

*"In marriage, you cannot divide responsibilities so that a man should play this part and a woman should play that part. What if you both work in an office and you both leave home at the same time. Would you say you would leave your wife behind so she alone would take care of the chores and get the children ready for school? If she's fired from work for being late constantly, whose fault would it be? It is your fault. So, the best thing is to perform these roles together...But if you should leave everything in the care of the woman, that would cause a lot of problems like separation." (Mixed FGD-#3)*.

Mothers perceived their role as main primary caregivers of children as they were the ones who spent the most time at home. However, a few caregivers mentioned that the father is the primary caregiver because he provides financially,

which allows the family to pay for the children's school, clothes, and food. One father shared: *"I am involved in raising the children, I give, I do everything, I give money, she has not given money before. Let's say that today the man does not have so she would step in and say get some money to go; no, she has not done that before mm hmm." (Father-IDI-#4).* Mothers also reported that they were responsible for household chores and taking care of the family. Some caregivers explained that they had not strictly divided their roles but rather believed in supporting one another and sharing tasks based on who is available. Indeed, although less commonly reported, a few fathers expressed gratitude and appreciation for their partners for contributing financially:

*"In my home, our roles are not divided. Sometimes when I go to work, by the time I come back it will be late. So, at times, I would call my husband and ask him if he can prepare something for dinner... If he doesn't have money to give to the children for school, I will use my own money to support because next time when he has it, he will do the needful. So, we support each other." (Mother-FGD-#5).*

**Positive parenting behaviours: Practices, facilitators, and barriers.** Many caregivers reported engaging in various learning activities with their child(ren) such as reading, singing, playing, and writing. Specific forms of play included "ampe" (a local game involving jumping and clapping) and playing educational games on the phone. Caregivers explained that these learning activities were impactful because children can remember and repeat what they have learned, which will help them be better prepared for school. In contrast, some caregivers reported that their low levels of education was a barrier to engaging in learning activities as they could not help their children read or write. They overcame this by asking their children about what they learned in school, drawing with them, or having someone else, such as a family member, tutor them. Additionally, caregivers shared that they taught children what is right and wrong to help them become good people:

*"I give them house training. And because I didn't go to school the house training my father taught me is what once a while I remember...like, uh huh, for instance like advising you not to join any bad company, joining gang in school." (Father-IDI-#6).*

One facilitator to engaging in positive parenting behaviours was having a positive couple's relationship. Caregivers explained that children will have a comfortable environment at home and be happy when their parents have a good relationship. They shared that having a good relationship sets an example for the children, shaping their values and behaviours. Furthermore, caregivers explained that harmony in the relationship and avoiding conflicts in front of the children encourages positive and respectful social interactions both inside and outside the home. They emphasized that if children are exposed to arguments, then they will be unhappy and learn to argue with others as well:

*"When you quarrel the children are seeing that you are quarrelling and that is what will be kept in the child's mind. Please anyone taking care of children, if you are home, insults and the like do not insult, when you insult the children are hearing, one day when they go out, the child will insult someone and the child will share he learnt that from his parent." (Nonparental caregiver-IDI-#7).*

Participating in religious and social gatherings was described as another facilitator to supporting positive parenting behaviours, particularly by respected community members. They explained that places such as churches and mosques allowed parents to learn how to raise children and improve their parenting. One community member shared, *"It's like the church does that in terms of programs and then when people come to church on Sunday, we try to educate them about most of the social challenges..." (Community member-IDI-#8).* These gatherings and programs help caregivers learn about child development, social issues, and mental health, which leads to more effective parenting. Awareness of appropriate

disciplinary methods was another facilitator to supporting positive parenting behaviours. Caregivers emphasized the importance of not using physical discipline on younger children as they would not be able to understand what they had done wrong, making the discipline ineffective. Caregivers also explained that they avoided using a cane or excessive force when using physical discipline to prevent making children sad, scared, or negatively impact their cognitive development. They emphasized that discipline should teach children about their mistakes and not to make them feel unsafe in their home. Some caregivers also discussed the importance of disciplining children soon after they have misbehaved and consistently for the discipline to be effective and so they do not confuse the child:

*"So, if you want to discipline, discipline the child immediately....let the child know what he or she is doing is wrong. But when you punish the child two or three days after he or she has done something wrong the child wouldn't know what he or she did, the child feels you are being wicked on him or her." (Father-FGD-#9).*

Finally, financial stability was discussed as a facilitator to engaging in positive parenting behaviours. Having a job and one that pays well allowed caregivers to better take care of their children because they were able to provide for their nutrition, education, and costs related to healthcare: *"When I go to work, I get some money which I bring home to cater for the home." (Nonparental caregiver-IDI-10103C).* Additionally, educational programs that taught caregivers about financial planning or a trade (e.g., soap-making, sewing, braiding, etc.) so they could start their own business were also beneficial: *"Sometimes we go beyond to talk about how they themselves can even make money, from petty-petty trading to support whatever the partner is bringing." (Community member-IDI-#10).*

In terms of barriers to engaging in positive parenting behaviours, many caregivers talked about limited time with children due to work or being tired. One mother shared, *"I don't really have enough time for them [children]. Because I need to do something to support their father in their upbringing." (Mother-IDI-#11).* This lack of time and energy made it difficult for caregivers to engage in learning activities, play, and other positive interactions that support their children's development. Caregivers often reported having to choose between quality time with their children and maintaining financial stability:

*"I would have to take a day off from my job and I can't take two days off, it should be a day off because the children are many. And you can't wake up and not make something meaningful in 24 hours unless I decided not to work...if I don't make anything meaningful the following day. I will have to take some of the money I saved for us to use, that means the money will be reducing." (Father-IDI-#6).*

Another barrier to engaging in positive parenting practices was the lack of paternal involvement.. Although a small number of fathers reported engaging in caregiving more equally, mothers, other female caregivers, and respected community members emphasized that this was not the norm. One pastor shared, *"The father's role on children within the community is quite very poor, very poor. I see more of the mothers actively pushing." (Community member-IDI-#12).* The lack of paternal involvement was described as not only putting more burden on mothers, but also hindering fathers' communication and bonding with their child:

*"I do remember, my daughter, my first born she wanted to tell us her aspiration but because the father doesn't stay at home she can't tell the father. So, she wrote it on a paper and posted in the room, when you enter the room, you will see it posted there, she wrote and pasted it on the wall." (Mixed-FGD-#13).*

Alcohol use was described as another barrier, with caregivers sharing that drinking too much negatively affected caregivers' ability to fulfill responsibilities, discipline children, and serve as a positive role model. Caregivers shared that it could also lead to children distancing themselves from their parents, losing respect for them, hearing negative things

neighbors say about their parents, and increasing their risk of drinking when they grew older. One father explained that drinking can also lead to marital conflicts:

*"As a family man, if you make it your habit to drink alcohol all the time, firstly, you cannot perform your responsibilities well as a father. Secondly, you cannot give your children the appropriate discipline and training you ought to give them because anytime you come home, you're fully drunk. So, the children cannot benefit or learn anything good from you. In some marriages, overuse of alcohol results in disrespect in the marriage because it affects our child training and responsibilities at home as well." (Mixed-FGD-#14).*

Finally, experiencing financial difficulties was a major barrier to engaging in positive parenting behaviours. Caregivers reported that not having work or enough money made it hard for them to provide for their children's nutrition, education, healthcare, and necessities. Experiencing financial strain made it difficult for caregivers to nurture their children, engage in stimulating activities, and allow time for bonding as they are busy working to provide: *"In my making time for them, it all relies on money, because if I do not have money, I will not be able to stay home and make time for them." (Father-IDI-#15).*

**Parental mental health: Caring practices, facilitators, and barriers.** Caregivers reported caring for their mental health in many ways including engaging in stress-relieving activities (e.g., listening to music, spending time by the beach, cooking, playing with their children, visiting friends), engaging in religious practices and relying on their faith (e.g., attending church or mosque, praying, and listening to sermons), and reaching out to friends, family, elders in the community, and religious leaders when experiencing stress. For example, believing in God or Allah during hard times gave caregivers hope, resilience, and helped them face life's challenges, which reduced their worries and feelings of despair: *"I try not to rely on anybody. I pray to God, sometimes he uses people to help me. That sometimes makes the sadness go away and I'm happy." (Nonparental caregiver-IDI-#16).* Additionally, talking to others helped caregivers forget about their worries and provided them comfort and practical support such as advice or money. The person caregivers sought support from depended on the issue that was worrying them. For example, caregivers explained that marital conflicts were only discussed with parents or elders in the community who had the same experiences. Others believed that one could talk to religious leaders about martial conflict, while others refrained from doing so for fear of making matters worse. One mother shared that when she is faced with a challenging and stressful situation, she likes to talk to her parents because they know her best and are able to advise her accordingly: *"They are my parents, they birthed me, so they know me better, they are able to tell me do this, do not do this, this will not help. Tackle things this way and the like." (Mother-IDI-#17).*

In terms of facilitators for maintaining good mental health, many caregivers talked about financial stability. Having access to stable employment enabled caregivers to meet their family's financial needs, reducing stress and anxiety: *"... But if I work and earn something, I feel happy that whenever any child of mine will face any problem, maybe in terms of an illness or the likes, I have something on me and can solve it for them." (Father-IDI-#18).* Overall, financial stability allowed caregivers to take care of their child without worrying about how to pay bills or make ends meet. Additionally, having a job gave caregivers a sense of purpose, accomplishment, and control over their lives. For instance, caregivers involved in local businesses like farming and artisanal work not only supported their families financially but also took pride in their work: *"Sometimes, when I sew a dress for a customer and she is happy, and after that she pays me what she owes, that makes me feel that she is satisfied with my work." (Mother-IDI-#11).*

One barrier to supporting parental mental health was social isolation. Without strong social support networks, caregivers described having fewer opportunities to share their struggles, seek advice, or receive practical help, making them feel alone. Some caregivers reported they did not have friends or family they could talk to about their worries because they had moved from another community and knew no one. Others did not believe in sharing their worries with others because they were afraid they would give incorrect advice: *"As for that one, I advise myself because it's not good to tell people*

*about your relationship issues, they might advise you wrongly. You should advise yourself and take the problem out of your mind." (Mother-IDI-#19).* In contrast, a small number of caregivers explained that they did not talk to others about their stresses because they did not want to be a bother or burden them with their worries, fearing that sharing their worries with others would drive them away:

> *"Let me say when I do not have money, it's only my brother I can tell or [name redacted], my best friend, but when we meet friends once a while, we will converse. If I tell them this is making me sad or I do not have money, they will see the kind of person I am: poor person every time he is disturbing us, let us forget him and go." (Father-IDI-#20)*

Lack of access to mental health services was another barrier to supporting parental mental health. Many caregivers reported that there were no resources in the community that helped them take care of their mental health: *"Oh, honestly speaking, as for this place, there is nothing like that that helps us in that manner [mental health]." (Father-IDI-#21).* Another barrier to supporting parental mental health, specific to mothers and other female caregivers, was the lack of support from their partners. Mothers and other female caregivers reported feeling stressed and overwhelmed when they did not receive help from their partners with household chores and childcare. They explained that sometimes they may take out their anger or frustration out on their children, which further worsened their mental health:

> *"What causes us stress, we women sometimes do all the house chores alone, for instance while you are cooking and washing simultaneously your husband will be in the room and not helping you. And when you don't get anyone to help you do it, you sometimes you displace your anger onto your children when they do something wrong. I normally will insult you, when my husband is in the room I am like, so can't you even take care of the young child so that I finish with the house chores. And when this happens, I get angry." (Mixed-FGD-#22).*

Similar to discussions about positive parenting behaviours, financial difficulties were also identified as a barrier to caring for one's mental health. Not having enough money made it difficult for caregivers to provide for their home and children, which led to stress. This was especially significant for fathers, who mostly described their primary family responsibility as providing financially:

> *"There are a lot of needs, that I can be up from morning, but I do not have money to use to take care of the family, then it becomes a major problem for me. It becomes a thinking problem for me. Then if for twenty-four hours, I do not get any help from anywhere, then this problem will be on my mind." (Father-IDI-#18).*

Additionally, some fathers reported that not having money could lead to arguments with their partners, while some mothers and female caregivers reported feeling worried when their partner did not have money for food or their child's schooling. This ongoing financial insecurity contributed to high levels of stress and created conflict between partners. One community member noted, *"When a man doesn't have money so as to take care of the family, the family isn't happy." (Community member-IDI-#23).* Overall, experiencing financial difficulties increased feelings of hopelessness, depression, and anxiety among caregivers. These feelings and worries were even expressed when they were asked about what gives them happiness in life. Instead of answering with what makes them happy, many parents talked about how they are always worried about not being able to provide for the family. For example, one father who was asked what gives him joy answered:

> *"Since this president came, my work is not going on well… payments can be delayed for as long as 2 to 3 months and maybe by then you need to pay your children's school fees or something…people used to come to the beach a lot during holidays but since the lockdown people don't come to the beach often, our personal jobs are now ruined." (Father-IDI-#20).*

Finally, alcohol use was described both as a strategy for coping with mental health difficulties and a barrier to good mental health. Some caregivers reported drinking alcohol to forget about stresses related to unemployment or conflicts at home, with some sharing that drinking was better than overthinking: *"So like my brother said, if drinking alcohol won't be a problem for you, you could drink a little and sleep, by the time you wake up, you'd be feeling ok. Otherwise overthinking these problems might just cause you more harm."* (Mixed-FGD-#24). However, most caregivers acknowledged that drinking alcohol to manage stress was not a useful coping strategy. They explained that it only helped them forget about their stress temporarily and did not help them address the root cause of the problem. Additionally, they reported that alcohol did not help them feel better like they thought it would: *"Me I will be honest, when my husband died because of pressure and what my family did I felt that when I take in alcohol it was going to help me solve...in doing that I realized that was not helping me, it made me feel not to get close to people anymore..."* (Nonparental caregiver-IDI-#25). Some caregivers also talked about how alcohol use could lead to conflicts between spouses. For example, some mothers expressed frustration with their partners for spending money on alcohol when did they not even have money for other necessities *"Please, the alcohol, it makes you weak, it makes you eat a lot and you also do not have money to buy food with, so it made it seem as if there was an argument coming between us, so I spoke at length with him and left him alone."* (Mother-IDI-#26).

**Existing community services: Experiences and interest in additional programs.** Caregivers reported that there were limited community services for parenting and parental mental health. The most frequently mentioned services were health services provided by mobile nurses and polyclinics which increased parents' knowledge on child health and helped them keep their children healthy, and Livelihood Empowerment Against Poverty (LEAP, a cash transfer program delivered by the Ministry of Gender, Children, and Social Protection), which provided money and financial education.

*"At times nurses are mobilized to come round the community to talk to us and educate us on how to take care of our children to avoid contracting diseases. There are also two polyclinics here where we go to whenever we feel sick."* (Mother-FGD-#27).

*"How it helps is that, let us use the LEAP program as an example, one that I was able to attend, I got help from there to help my child to go to school. Although it is not big but once that money comes, I make sure I put it down for that child. So, when times are getting difficult, that is also helping the child to go to school little by little...also, we get advice from the personnel that come, that...you cannot depend on one source of income, they really advise us very well..."* (Father-IDI-#28).

Outside of these formal services, caregivers described childcare support from family (e.g., older children, grandparents, aunts, and uncles) and schools as useful resources, sharing that the former allowed them to focus on other tasks or give them a break from childcare and the latter provided children with education and taught important values to help them grow into good people.

*"Like the children in our house, they help me to take care of the child. Maybe as I am doing something, they come and take hold of him/her. With that also, they come and help. Apart from that there is nothing else... I feel relaxed. I become free."* (Mother-IDI-#26).

*"Because there are a lot of children that when you see them, you notice that they do not have training, or these people have not had any teachings. Ehehh, but with the school my children attend, there are a lot of teachings that when they come home, they do them..."* (Father-IDI-#18).

Caregivers expressed interest in attending meetings to talk about parenting and parental mental health, highlighting the need for more structured and accessible services. For parenting, they shared that this would be a great opportunity to learn everything about raising children (their education, feeding, and health care) and how to take care of their family

and promote positive relationships with one another: *"I want to learn a lot more about how to raise my children very well and how to cater for my family and partner and how to live well with everyone. Whatever they'll teach I'm willing to learn."* (Mother-IDI-#11). Caregivers were also interested in learning a trade or about employment opportunities to provide for their family, which would help them engage in positive parenting practices: *"If it is about learning a trade or establishing a business, I would like it so that I can also use that to support myself and the children."* (Nonparental caregiver-IDI-#29). They expressed that they want to learn about these topics to help their children have a good future and become good people.

When asked about topics they would like to learn about regarding mental health, caregivers expressed interest in learning about strategies for managing stress, tiredness, sadness, as well as learning how to be more patient: *"For instance, when I attend these meetings I hope they teach us so many things about stress so that whenever I am home and such thing happens to me I know how to manage it because I've had education on that, uh huh."* (Mother-IDI-#30). Some caregivers emphasized the need for a space to discuss their mental health and learn how to support each other.

Participants highlighted several factors that could make the intervention successful. First, participants explained that meetings would have to take place at a date and time that most caregivers would be available. Evening sessions were thought to be the best time to accommodate caregivers, to avoid conflicts with their daily routines and responsibilities. However, many caregivers, especially fathers, shared that they might not be able to attend due to work. For this reason, participants emphasized that the meetings should take place outside of working hours:

> *"We live in a fishing community and Tuesdays are marked for not going fishing. And for those of us who do white collar jobs, we are only available on weekends. So, we would urge you to give us a notice in advance, if you should choose Tuesday or Saturday, inform us in advance, that will help us to schedule our time appropriately."* (Father-FGD-#31).

Participants shared that making announcements (via community centers, radio stations, and word of mouth) could also help with the success of the intervention by ensuring that many caregivers will know about the program and plan accordingly to attend: *"Umm firstly, if you will come here, you have to make announcements, you have to announce it, you see, then when you get a month that is suitable you take it. There is a place for making announcements here."* (Community member-IDI-#32). They also emphasized the importance of involving community leaders for the intervention's success, emphasizing that while caregivers would be interested in attending these meeting, seeing community leaders support the meetings would further encourage their participation: *"Yes, it depends on how you send the message, who gives the message and where you give the message. Always involve community leaders."* (Community member-IDI-#8). Finally, the inclusion of incentives (e.g., food or money) to encourage participation and ensuring the accessibility of the intervention (both in terms of physical access by covering transportation costs and use of clear and simple language) were highlighted as key strategies to facilitate its success.

## Discussion

This formative study sheds light on how caregivers in southern Ghana care for young children and their own mental health by providing new insights into how socio-economic and cultural conditions shape parenting and parental mental health. Participants provided rich descriptions of how caregivers engage in parenting practices and support their mental health, the barriers and facilitate they face in doing so, and their experiences of existing community services to support parenting and parental mental health. Findings reveal that financial insecurity, rigid gender norms, and the absence of accessible mental health services undermine both caregiver well-being and child development. Our results highlight the importance of addressing structural and relational challenges to ensure that multi-component parenting and mental health interventions are contextually relevant, gender-responsive, and address local needs.

Echoing previous research in Ghana and other African contexts, financial instability emerged as a central barrier to both positive parenting and mental health [25,30–32]. This stress was especially acute among fathers, whose culturally

sanctioned role as sole providers placed them under immense pressure. Many expressed anxiety when unable to fulfil this expectation, revealing how economic hardship not only undermines paternal engagement but also strains mental well-being. Notably, some fathers valued and welcomed their partners' financial contributions, an evolving stance possibly driven by post-pandemic economic realities and shifting gender norms [33,34].

We found that financial stress was a key driver of parental distress and limited engagement with children, particularly among fathers. This suggests that interventions that incorporate economic strengthening, such as cash transfers, may enhance both participation and impact. This aligns with caregivers' own suggestions in our study, where several expressed interest in learning trades and receiving financial literacy training to better support their families. Participants also reported that financial stability not only improved material conditions but moreover, contributed to caregiver self-efficacy and reduced psychological distress. These factors are essential to sustaining positive parenting behaviours, further highlighting the need for economic support. There is evidence that integrating cash transfers into parenting interventions can improve both child and caregiver outcomes. For example, a study in Tanzania found that integrating an economic strengthening component into a parenting intervention compared to the control, parenting-only, and economic strengthening-only groups led to decreases in maltreatment, support for corporal punishment, and child behaviour problems [35]. Furthermore, cash transfer interventions across sub-Saharan Africa have reported a decrease in maternal stress and improvements in economic well-being [36,37]. These findings suggest that addressing financial strain not only supports material conditions but also alleviates psychological distress and promotes positive parent-child relationships. Relatedly, this tension between financial provision and direct caregiving underscores the need for interventions that acknowledge caregivers' time constraints and offer feasible, low-burden strategies for engagement.

Paternal disengagement also emerged as a significant barrier to positive parenting and maternal well-being. According to participants, this disengagement was driven by time constraints and entrenched gender norms, which left mothers with disproportionate household chores that increased maternal stress. This stress sometimes resulted in displaced frustration towards their children, highlighting how gendered roles can negatively affect both mental health and parenting practices. Addressing gendered divisions of labour is therefore central not only to paternal engagement but also to reducing caregiver stress. Some fathers in our study recognized the importance of shared caregiving and voiced frustration at community stigma around men performing domestic roles. This suggests that some community members may be open to engaging in norm-shifting conversations. Evidence shows that interventions with gender-transformative approaches lead to more equitable division of responsibilities and paternal engagement [27,38–40]. This is crucial for improving ECD. For example, the *Moments That Matter* program implemented in rural Kenya and Zambia used a community-led approach consisting of home visits and activities centering on gender-equitable parenting [41]. The intervention led to an increase in fathers' participation in household chores and stimulating interactions with their children, as well as an improvement in their attitudes on gender-equal parenting [41]. These findings highlight the potential of gender-transformative strategies to target rigid gender norms and improve child and family outcomes. To be effective, gender-transformative interventions must be adapted to local realities and leverage the willingness of some fathers to challenge traditional roles when provided with safe, supportive spaces.

Finally, our findings also showed that there was a lack of mental health services in the community, making it difficult for caregivers to support their mental health. Caregivers repeatedly emphasized the near absence of local mental health resources, noting that even basic support, such as someone to talk to, was lacking. This mirrors national trends of limited decentralized mental health infrastructure in Ghana [42]. Without accessible services offering psychosocial support and practical strategies for managing distress, caregiver well-being is likely to deteriorate [43,44]. Several participants expressed interest in community-based discussion groups or sessions focused on addressing stress, sadness, and emotional support. They also suggested that even low-intensity interventions delivered through existing platforms could be meaningful. Adapting and integrating global models such as WHO's Thinking Healthy Program or UNICEF and WHO's Caring for the Caregiver approach into existing local health infrastructure, while ensuring cultural and linguistic relevance,

could be a feasible pathway forward [44–46]. Furthermore, given the role of religious and social spaces in supporting parenting practices and emotional resilience, future interventions might benefit from collaborating with local faith institutions to enhance reach and trust [25,26,41].

This study provides novel data on caregivers' and community members' perspectives on how caregivers care for their young children and their own mental health in a Ghanaian context. It also sheds light on the barriers and facilitators that caregivers face when engaging in these behaviours. A key strength of this study lies in its inclusion of diverse caregiving voices, from mothers, fathers, nonparental caregivers, and community members, allowing for a multidimensional understanding of parenting and mental health in low-resource, coastal communities. However, the predominance of fishing as a livelihood in the study area may limit applicability to contexts where fathers are more consistently present in the home. Our use of purposive sampling may also limit the relevance of findings for other populations. Future research could explore whether these patterns persist in urban or agricultural regions. Furthermore, although the combination of IDIs and FGDs provided both individual and collective perspectives on sensitive topics, it is possible that social desirability bias was present and participants may not have felt comfortable discussing some topics (e.g., intimate partner violence), which were thus underreported. Finally, while the sample focused on two-parent households, the inclusion of extended kin caregivers offers some insight into alternative caregiving arrangements common in Ghana.

## Conclusion

This formative qualitative study explored how parents, caregivers, and community members in southern Ghana experience and navigate parenting and mental health difficulties. Findings underscore the need for a contextually-relevant multi-component parenting and parental mental health intervention that combines economic strengthening with gender-transformative and community-based mental health support. Researchers, policymakers, and health providers need to work alongside communities to integrate these findings into the development of policies and interventions to better support Ghanaian families.

## Supporting information

**S1 File. COREQ checklist.**
(PDF)

**S2 File. PLoS Inclusivity in global research questionnaire.**
(DOCX)

## Acknowledgments

The study authors would like to express our appreciation to the study team for collecting the data and to study participants for their time and contributions. Marilyn N. Ahun was supported by the Victoria S. Levin Award for Early Career Success in Young Children's Mental Health Research from the Society for Research in Child Development.

## Author contributions

**Conceptualization:** Richard Appiah, Marilyn Naana Ahun.

**Data curation:** David D. Kotey, Marilyn Naana Ahun.

**Formal analysis:** Faiza Abdul, Kafui K. Dzorgbesi, Shirley-Anne Lutterodt, Marilyn Naana Ahun.

**Funding acquisition:** Marilyn Naana Ahun.

**Supervision:** Richard Appiah, Marilyn Naana Ahun.

**Writing – original draft:** Faiza Abdul, David D. Kotey, Marilyn Naana Ahun.

**Writing – review & editing:** Faiza Abdul, Kafui K. Dzorgbesi, Shirley-Anne Lutterodt, David D. Kotey, Richard Appiah, Marilyn Naana Ahun.

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
