## [Decision Letter · Decision Letter 0]

8 Sep 2025

PGPH-D-25-01851

Formative research to inform the design of a multi-component parenting and parental mental health intervention in Ghana

Dear Dr. Ahun,

Thank you for submitting your manuscript to PLOS Global Public Health. After careful consideration, we feel that it has merit but does not fully meet PLOS Global Public Health’s publication criteria as it currently stands. Therefore, we invite you to submit a revised version of the manuscript that addresses the points raised during the review process.

We look forward to receiving your revised manuscript.

Kind regards,

Bibhav Acharya

Academic Editor

Journal Requirements:

2. Please ensure that your Ethics Statement is available in its entirety at the beginning of your Methods section, under a subheading 'Ethics Statement'.

4. For studies involving third-party data, we encourage authors to share any data specific to their analyses that they can legally distribute. PLOS recognizes, however, that authors may be using third-party data they do not have the rights to share. When third-party data cannot be publicly shared, authors must provide all information necessary for interested researchers to apply to gain access to the data. (https://journals.plos.org/plosone/s/data-availability#loc-acceptable-data-access-restrictions)

Additional Editor Comments (if provided):

Reviewer #1:

This is a detailed and well-written manuscript. It explores the link between parenting and parental mental health. Given the post-COVID socio-economic turbulence which has significantly affected families and caregiving in LMICs, this study is timely, important and offers meaningful insights. The inclusion of diverse caregivers is positive.

Areas to note- 1.Title alignment - consider refining the title to better reflect the findings and contributions to make it more outcome-driven and results-oriented.

2. Key-words - Look at the keywords again, the manuscript contains more powerful keywords. eg "mental health" consider "parental mental health" focusing on parents' psychological well-being covered in the manuscript.

3. Introduction-This section needs stronger comparison with existing global data.

4. Methods - COREQ (provide reference)

5. Sampling -i. Clarify whether mobilizers followed a standardized recruitment guide to avoid biases. ii. Clarify how you ensured diversity and equal representation of voices. iii. Consider a brief explanation of why 6-60months range was chosen.

6.The discussion session is wordy. Shorten complex sentences to improve readability.

Overall, it is a strong piece of work, these minor revisions will enhance rigorousness.

Reviewer #2:

(In attachment)

Reviewers' comments:

Reviewer's Responses to Questions

**Comments to the Author**

1. Does this manuscript meet PLOS Global Public Health’s publication criteria?

Reviewer #1: Yes

Reviewer #2: Yes

2. Has the statistical analysis been performed appropriately and rigorously?

Reviewer #1: Yes

Reviewer #2: N/A

3. Have the authors made all data underlying the findings in their manuscript fully available (please refer to the Data Availability Statement at the start of the manuscript PDF file)?

Reviewer #1: Yes

Reviewer #2: Yes

4. Is the manuscript presented in an intelligible fashion and written in standard English?

Reviewer #1: Yes

Reviewer #2: Yes

Reviewer #1: This is a detailed and well-written manuscript. It explores the link between parenting and parental mental health. Given the post-COVID socio-economic turbulence which has significantly affected families and caregiving in LMICs, this study is timely, important and offers meaningful insights. The inclusion of diverse caregivers is positive.

Areas to note- 1.Title alignment - consider refining the title to better reflect the findings and contributions to make it more outcome-driven and results-oriented.

2. Key-words - Look at the keywords again, the manuscript contains more powerful keywords. eg "mental health" consider "parental mental health" focusing on parents' psychological well-being covered in the manuscript.

3. Introduction-This section needs stronger comparison with existing global data.

4. Methods - COREQ (provide reference)

5. Sampling -i. Clarify whether mobilizers followed a standardized recruitment guide to avoid biases. ii. Clarify how you ensured diversity and equal representation of voices. iii. Consider a brief explanation of why 6-60months range was chosen.

6.The discussion session is wordy. Shorten complex sentences to improve readability.

Overall, it is a strong piece of work, these minor revisions will enhance rigorousness.

Reviewer #2: The reviewer comments has been attached via the orchid account. Overall, the study was well done covering a very insightful aspect of parenthood this time their mental health which is essential for their wellbeing and that of the children.

**Do you want your identity to be public for this peer review?** For information about this choice, including consent withdrawal, please see our Privacy Policy

Reviewer #1: **Yes:** Elizabeth Korasare

Reviewer #2: No

---

## [Editor Report · Decision Letter 1]

7 Nov 2025

PGPH-D-25-01851R1

Community-identified priorities for parenting and parental mental health in coastal Ghana: Formative insights to shape a multi-component intervention

Dear Dr. Ahun,

Thank you for submitting your manuscript to PLOS Global Public Health. After careful consideration, we feel that it has merit but does not fully meet PLOS Global Public Health’s publication criteria as it currently stands. Therefore, we invite you to submit a revised version of the manuscript that addresses the points raised during the review process.

We look forward to receiving your revised manuscript.

Kind regards,

Bibhav Acharya

Academic Editor

Journal Requirements:

Additional Editor Comments (if provided):

Thank you for your thorough response to the reviewers' comments.

One issue is that you will need to remove all citations associated with a reference that is under review, as those are not allowed (Ahun et al., under review). If including this information is essential, you may upload the relevant data as a supplement and refer to that. Otherwise, I suggest that you simply remove.
---

## [Editor Report · Decision Letter 2]

15 Jan 2026

Community-identified priorities for parenting and parental mental health in coastal Ghana: Formative insights to shape a multi-component intervention

PGPH-D-25-01851R2

Dear Dr. Ahun,

We are pleased to inform you that your manuscript 'Community-identified priorities for parenting and parental mental health in coastal Ghana: Formative insights to shape a multi-component intervention' has been provisionally accepted for publication in PLOS Global Public Health.

Best regards,

Bibhav Acharya

Academic Editor